# Fine Mapping of Stripe-Rust-Resistance Gene *YrJ22* in Common Wheat by BSR-Seq and MutMap-Based Sequencing

**DOI:** 10.3390/plants11233244

**Published:** 2022-11-25

**Authors:** Can Chen, Weihao Hao, Jingchun Wu, Hongqi Si, Xianchun Xia, Chuanxi Ma

**Affiliations:** 1Laboratory of Wheat Breeding, College of Agronomy, Anhui Agricultural University, Hefei 230036, China; 2Key Laboratory of Wheat Biology and Genetic Improvement on Southern Yellow & Huai River Valley, Ministry of Agriculture and Rural Affairs, Hefei 230036, China; 3National United Engineering Laboratory for Crop Stress Resistance Breeding, Hefei 230036, China; 4Institute of Crop Sciences, National Wheat Improvement Centre, Chinese Academy of Agricultural Sciences (CAAS), 12 Zhongguancun South Street, Beijing 100081, China; 5Anhui Key Laboratory of Crop Biology, Anhui Agricultural University, Hefei 230036, China

**Keywords:** BSR-Seq, MutMap-based cloning, *Puccinia striiformis tritici*, *Triticum aestivum*

## Abstract

Identification and accurate mapping of new resistance genes are essential for gene pyramiding in wheat breeding. The *YrJ22* gene is a dominant stripe-rust-resistance gene located at the distal end of chromosome 2AL, which was identified in a leading Chinese-wheat variety, Jimai 22, showing high resistance to CYR32, a prevalent race of *Puccinia striiformis tritici* (*Pst*) in China. In the current study, 15 F_1_ and 2273 F_2_ plants derived from the cross of Jimai 22/Avocet S were used for the fine-mapping of *YrJ22*. The RNA-Seq of resistant and susceptible bulks of F_2_ plants (designated BSR-Seq) identified 10 single-nucleotide polymorphisms (SNP) in a 12.09 Mb physical interval on chromosome 2AL. A total of 1022 EMS-induced M_3_ lines of Jimai 22 were screened, to identify susceptible mutants for MutMap analysis. Four CAPS markers were developed from SNPs identified using BSR-Seq and MutMap. A linkage map for *YrJ22* was constructed with 11 CAPS/STS and three SSR markers. *YrJ22* was located at a 0.9 cM genetic interval flanked by markers H736 and H400, corresponding to a 340.46 kb physical region (768.7–769.0 Mb), including 13 high-confidence genes based on the Chinese Spring reference genome. *TraesCS2A01G573200* is a potential candidate-gene, according to linkage and quantitative real-time PCR (qPCR) analyses. The CAPS marker H732 designed from an SNP in *TraesCS2A01G573200* co-segregated with *YrJ22.* These results provide a useful stripe-rust-resistance gene and molecular markers for marker-assisted selection in wheat breeding and for further cloning of the gene.

## 1. Introduction

Wheat (*Triticum aestivum*) is a widely grown momentous food-crop, with an estimated global production of 776.7 million tonnes in 2020 [1]. Stripe rust from *Puccinia striiformis* f. sp. *tritici* (*Pst*) is a very serious fungal disease, which affects wheat-grain yield and quality and brings great challenges to wheat production worldwide [2]. A large number of *Pst* spores accumulated on leaves strongly weaken the photosynthetic capacity, eventually leading to yield reduction [3,4]. Due to wide spreading and difficult prevention and control, stripe rust has been ranked as a first-grade national crop disease by the Ministry of Agriculture and Rural Affairs of China [5]. Stripe rust occurred on over four million ha in China in 2020 [6]. The utilization of resistant cultivars remains the most economical, environment-friendly and effective way to control the disease. At present, 84 stripe-rust-resistance genes have been formally catalogued [7,8,9,10,11,12,13], but relatively few genes maintain resistance to the predominant races CYR32, CYR33 and CYR34 in China [14,15], which leads directly to the loss of resistance to stripe rust in many wheat varieties. It is, therefore, very urgent to explore new genes and effective markers, to breed resistant wheat-varieties [16].

Molecular marker-assisted selection (MAS) is effective in the pyramiding of resistance genes [17]. Developing diagnostic or tightly linked molecular markers is essential for MAS. However, the large and complex wheat genome largely hinders the fine mapping and cloning of genes and the development of molecular markers [18]. Although many wheat resistance-genes have been identified, most of them are mapped at large intervals. To date, nine wheat stripe-rust resistance-genes have been cloned, including the recently cloned *Yr28/YrAS2388* [19,20]. The *Yr5, Yr7* and *YrSP* genes were identified by mutant resistance gene enrichment sequencing (MutRenSeq), while the others were isolated by map-based cloning [21,22,23,24,25]. Among them, three are adult plant resistance (APR) genes, including *Yr18*, encoding an ABC transporter gene, *Yr36*, a kinase-START gene, and *Yr46*, encoding a hexose transporter. The other cloned *Yr* genes show all-stage resistance (ASR), encoding a nucleotide-binding site (NBS) and leucine-rich repeat (LRR) proteins, except for *Yr15* [26].

The rapid development of molecular technologies, such as transcriptome and resequencing, greatly accelerates the pace of resistance-gene localization and map-based cloning [27]. With the development of next-regeneration sequencing (NGS) assisted pooling [28], the BSR-Seq that combines bulked segregant analysis (BSA) and RNA-Seq has been widely used in the fine mapping and cloning of genes in wheat [29]. Using this method, more than a dozen disease resistance genes and dwarf genes have been fine mapped, such as *PmQ*, *YrZH22* and *Rht-2BL*, indicating the effectiveness of BSR-Seq [30,31,32]. Many single-nucleotide polymorphisms (SNPs) excavated from RNA-Seq provide molecular markers for the fine mapping and map-based cloning of genes [33].

MutMap-based cloning comprises ethyl-methane sulfonate (EMS) mutagenesis, sequencing, and mapping [34]. Mutational Mapping (MutMap) is a rapid method for gene isolation, based on whole-genome resequencing, genetic mapping and causative mutations. Typically, mutations are mapped in an F_2_ population derived from a cross between the isolated-mutant line and a wild type, using a BSA-based approach [35]. It could accelerate the identification of the genomic positions of genes in rice by comparing and sequencing bulked DNAs from the F_2_ population from the cross of the mutant and wild-type parental line [36]. It has been applied in the identification of *OsCAO1*, which most probably caused pale green leaves and the semi-dwarf, using seven mutants of a Japanese rice cultivar Hitomebore [37]. Another strategy that is more straightforward for mapping mutations would be the comparison of the genome sequences of the mutant and wild type. Nordström performed a mapping approach by directly sequencing allelic mutants without crossing [38]. The causal genes were identified in rice and *Arabis alpina* by comparing *k*-mers in whole-genome sequencing. In model-plant species with small genomes, mapping by causative mutations and whole-genome sequencing has proven an efficient approach for identifying genes related to growth habit, color of leaves and blast resistance [39,40]. For example, *Ms1*, an essential gene for microgametogenesis in hexaploid wheat, was further mapped and cloned with 676 *msle* plants by MutMap-based cloning [41]. Another example related to resistance is the identification of candidate genes for *Fusarium* wilt and sterility mosaic disease in pigeonpea (*Cajanus cajan*), by a combined approach of sequencing-based bulked segregant analysis and whole-genome resequencing-based nonsynonymous SNPs [42]. This technique, in addition to the complete genome sequence of the Chinese Spring (CS) genome released by The International Wheat Genome Sequencing Project in 2017, will greatly accelerate the mapping and cloning of wheat genes on adapted and wild wheat plants [41,43].

Jimai 22, a facultative medium-gluten wheat for noodles and steamed bread, is a widely grown cultivar with the largest planting area during recent decades, and is an extensively used breeding parent in China, due to its high and stable yield, high adaptability, and resistance to multiple diseases [44]. It contains a new stripe-rust-resistance gene, *YrJ22*, against CYR32 mapped by SSR markers and 90 K SNP chip [45]. *YrJ22* was located at the distal end of chromosome 2AL in an 8.3 cM chromosome region between an SSR marker *Xwmc658* and SNP marker IWA1348. In the present study, we fine-mapped *YrJ22* to a close interval, using BSR-Seq and MutMap analysis, through the markers developed according to the variations of sequence analysis.

## 2. Results

### 2.1. Phenotypic Characterization of Stripe-Rust Resistance

Among the Jimai 22 × Avocet S F_2_ population, 1720 plants were resistant, whereas 553 were susceptible (Table 1), fitting a 3:1 segregation ratio based on a chi-square test (χ^2^ = 0.46, *P_1df_* > 0.05), and indicating that a dominant gene in Jimai 22 conferred resistance to stripe rust. Among 1022 lines from the EMS treatment, 990 were homozygous resistant and 27 segregated, whereas 5 were homozygous susceptible. In total, 34 plants from 21 different M_3_ lines were evaluated as highly susceptible, with IT = 4.

### 2.2. BSR-Seq and MutMap Analysis

Two RNA bulks were prepared by mixing RNA equally from 50 highly resistant (IT = 0;) F2 plants and 50 highly susceptible (IT = 4) ones. BSR-Seq genotyping was applied to detect various SNPs between the resistant and susceptible bulks. The BSR-Seq generated 132,101,678 and 178,459,594 reads (clean data) in the resistant and susceptible bulks, respectively. After quality control, we obtained 116,892,150 (88.49%) and 158,409,984 (88.77%) uniquely mapped reads for resistant and susceptible bulks, respectively. By calculating the ΔSNP-index values for both bulks, using the coordinates of uniquely aligned reads, 61 SNPs associated with stripe-rust resistance were obtained, based on the frequency distributions of genes with ΔSNP-index ≥ 0.9 on each chromosome (Figure 1), in which half were located on group 2 homoeologous groups and 10 were located on chromosome 2A, in a 12.09 Mb physical interval (761,299,381 to 773,385,650 bp).

After MutMap sequencing and data analysis, 618.27 Gb clean reads were obtained, with Q30 reaching 80%. Approximately 36,586,593 SNPs and 3,228,052 indels were subjected to a following mutation analysis after mapping to the reference genome using BWA. In total, 9,529,169 polymorphic SNPs between Jimai 22 and the susceptible mutant pool were detected, locating on 21 wheat chromosomes (Figure 2). In line with the locations of the SNPs, compared with the reference genome from SnpEff, it is possible to annotate whether the SNP loci are present in the inter-gene region, gene region or CDS region, and whether there is a code shift mutation (Appendix A). Among 40,852 SNPs with a non-synonymous substitute, 11,631 were located on homoeologous group 2 chromosomes (2A:1256, 2B:1645, 2D:8730). SNPs located in the coding region with non-synonymous substitution were found for the associations with the resistance phenotype. Through the locational and functional analysis of these SNPs, a 40.08 Mb region on chromosome 2A (732,984,789–773,060,805 bp) was identified, where 11 SNPs were assumed as potential candidates for causing the mutation, according to the SNP-index ≥ 0.9.

### 2.3. Collinearity Analysis of 2A, 2B, and 2D

Since the regions enriched by polymorphic SNPs from BSR-Seq and MutMap were not only chromosome 2A, we carried out a collinear sequence analysis of the target regions on chromosomes 2A, 2B, and 2D. The results showed that the 2A-, 2B-, and 2D-associated regions had high homology and collinearity (Appendix A). After analyzing the collinearity of the chromosomes, the different SNP-enriched regions on 2B (754,863,037 to 793,413,063 bp) from BSR-Seq, and on 2D (618,246,861 to 651,700,214 bp) from MutMap are highly homologous with the mapped bin (760,565,802–774,217,712 bp) of *YrJ22* in a previous study [41]. Finally, a 10.39 Mb region on 2A (762,567,882 to 772,967,287 bp) including 208 genes was identified. Eleven and five polymorphic SNPs in this region-mining from BSR-Seq and MutMap, respectively, were candidates for marker development.

### 2.4. Development of Polymorphic Markers

SNPs associated with stripe-rust resistance identified by BSR-Seq and MutMap analysis were selected for the development of CAPS markers. Three markers, H796, H896 and H034 were converted from SNPs of BSR-Seq, and H690 was developed from MutMap analysis. The flanking sequences of SNPs identified from BSR-Seq and MutMap analysis were used to blast the CS RefSeq, to design PCR primers. A linkage map was constructed with these SNP-CAPS markers first, while *YrJ22* was flanked in a 6.8 cM region by H896 and H796. Using functional annotation analysis, a total of 168 genes were screened in this genomic interval, including seven resistance (*R*) genes. After the sequence alignment of these genes between Jimai 22 and Avocet S in the 6.8 cM region, markers H400, H727, H732, H736 and H801 were developed using the SNPs and indels of these genes (Table 2). The markers were verified, based on polymorphisms between parental lines, and between resistant and susceptible DNA bulks (Figure 3).

### 2.5. Linkage-Map Construction and Candidate-Genes Analysis

Based on the genotypes and phenotypes of 553 susceptible F_2_ plants, a linkage map was constructed with newly developed and former linkage-markers (Figure 4). The interval for *YrJ22* was narrowed down from 8.3 cM to 0.9 cM by flanking markers, corresponding to a 340.46 kb physical region on chromosome 2AL, referring to CS RefSeq. The new genetic-interval for *YrJ22* overlapped with the previous study (Figure 5a). The SNPs in the genetic map were used to blast the Aikang58, Fielder, ArinaLrFor, Julius, LongReach_Lancer, Mace, SY_Mattis, Norin61 and CS RefSeq, to identify homologous contigs and gene annotations. The annotation of the 340.46 kb-region sequence identified 13 high-confidence genes (Table 3) [46]. To identify the candidate resistance-genes, all the 13 genes annotated in the target area were cloned from Jimai 22 and Avocet S for sequence alignment. Nine of them show polymorphisms between parents (Figure 5b). Among them, *TraesCS2A01G572900, TraesCS2A01G573000, TraesCS2A01G5740100* and *TraesCS2A01G573100* cannot be amplified in Jimai 22 and resistant F_2_ plants, but the full length of them from Avocet S were obtained and were highly consistent with the sequences in CS RefSeq by BLAST (identifying over 98.50%), indicating that there are great differences or indels between Jimai 22 and CS RefSeq (Figure 5c). There were several SNPs found by sequence alignment between Jimai 22 and Avocet S in *TraesCS2A01G5740000, TraesCS2A01G572700, TraesCS2A01G573200* and *TraesCS2A01G573600*. No difference was identified in four other genes after comparison, showing that it is not the candidate gene. The co-segregation marker H732 was developed from an SNP in *TraesCS2A01G573200,* the annotated gene in the target genomic-region (Figure 5d). The qPCR analysis demonstrated that expression of *TraesCS2A01G573200* in Jimai 22 was significantly higher than the susceptible line Avocet S at 12 and 24 hai with race CYR32 (*p* < 0.01) (Figure 6), indicating that *TraesCS2A01G573200* is a potential candidate gene for *YrJ22.*

## 3. Discussion

The use of resistance genes is one of the most effective and environment-friendly strategies for controlling disease. Durable and broad-spectrum resistance can be achieved in wheat by combining several resistance genes [47]. Fine-mapping and cloning of *Yr* genes provide a useful gene resource and molecular markers, to reduce the risks of stripe rust [48].

### 3.1. Combining Mutmap and BSR-Seq Is an Effective Approach to Fine-Mapping

To develop closely linked markers and to clone genes, map-based cloning is a classic method, while BSR-Seq and MutMap are gradually being used in map-based cloning in wheat [49]. BSR-Seq has been frequently used for the fine-mapping of resistance genes, such as *Yr15*, *Yr26,* and *PmQ* [21,31,50]. With the comparisons between resistant and susceptible bulks, SNPs linked with the target gene can be identified for development of molecular markers. In this study, 61 SNPs associated with the target gene by BSR-Seq were identified, ten, nineteen and one were located on chromosome 2A, 2B and 2D, respectively. Collinear sequence analysis showed that the target region is highly homologous among 2A, 2B and 2D. Wu found that high sequence-similarities among homeologs and homologs are major causes of false SNP-calling in common wheat [4]. In this situation, collinear sequence analysis can increase the accurate chromosomal localization. The mapping of *Yr26* showed that the genetic backgrounds of the cross parents and Chinese Spring affect the positioning results from BSR-Seq [50]. In this study, 2A is not the chromosome ranked top by the number of polymorphic SNPs from BSR-Seq or MutMap. To refine the chromosomal location and discriminate subgroups among three sub-genomes, chromosome-specific markers with physical information should be considered. This fully demonstrates that even with a huge and complex wheat genome, BSR-Seq and MutMap could help gene localization in specific homoeologous groups, with the help of chromosome-specific markers to assist fine-mapping rapidly.

### 3.2. Genetic and Physical Mapping of YrJ22

Jimai 22 has been a leading cultivar in China since 2006, and is also used as a founder parent for breeding, due to its immune reaction to CYR32 and excellent agronomic traits [45,51]. The stripe-rust-resistance gene, *YrJ22,* in Jimai 22 was mapped within an 8.3 cM interval on the distal region of 2AL bin 1–0.85–1.00 [45], while it was mapped in a 0.9 cM interval flanked by H400 and H736 corresponding to 0.3 Mb in the present study, in line with the previous result. Unlike *YrJ22*, *Yr1* and *Yr32* are two officially named *Yr* genes on 2AL [48]. Both differ from *YrJ2*, based on gene-specific PCR fragments and multi-pathotype tests [45]. Besides these, some temporarily named *Yr* genes were located on 2AL. *YrZM175*, the only temporarily named ASR gene on 2AL, from Zhongmai 175 was linked with markers YrZM175-CAPS1 and YrZM175-InD1, spanning a 636.4 kb genomic region [52]. Based on its physical position and linked markers, *YrJ22* appears to be located at a unique locus. Many genes for APR were previously mapped on 2AL, such as *Qyr.saas-2A* from Chuanmai 55, *QYr.cau-2AL* from Hongqimai, *QYr.caas-2AL* in Zhong 892, *QYrqin.nwafu-2AL* in QN142 and *QYr.wpg.6* in PNW, which is different from *YrJ22* according to the physical locations and characteristic reactions to *Pst* [53,54,55,56,57].

### 3.3. Analysis of the Candidate Region of YrJ22

Genes which translate into NLR proteins are major components of the innate immune system, and are often causal genes for disease resistance in plants [58]. Redundancy analysis indicates that ~30% of the NLR signatures are shared across all genomes in wheat cultivars [59,60]. The cloned ASR *Yr* genes, such as *Yr5* and *Yr10*, encode for NBS and several LRRs proteins. Because of the release of the CS sequence, some cloned *R* genes are not present in the wheat reference-genome [61]. In the present study, two out of thirteen genes in the candidate region were blasted with the NLR protein structure from CS RefSeq v1.0 as reference. Li cloned *Pm41* and found a presence/absence variation (PAV) in the *Pm41* locus between wild emmer wheat (WEW) accession IW2 and hexaploid wheat CS [62]. *Yr36* originated from WEW, was present in southern WEW populations and was absent in other WEW populations and in durum, and common wheat [63]. PAV in a gene encodes a coiled coil (CC), nucleotide-binding site (NBS), and leucine-rich repeat (LRR). The CNL protein was reported in the leaf-rust-resistance gene *Lr10* in hexaploid wheat, and is not closely related to other wheat *Lr* genes [64]. The sequence alignment of *TraesCS2A01G572900, TraesCS2A01G573000* and *TraesCS2A01G573100* indicates that the 252.76 kb (chr2A:768,729,058 to 768,981,815 bp) genomic sequence could be a PAV in the *YrJ22* locus between Jimai 22 and the reference genome. In *TraesCS2A01G573200,* we found some sequence variations, including 14 SNPs between Jimai 22 and Avocet S; three variations caused amino-acid substitution (Appendix A). A co-segregation marker, H732, was developed, based on the T-C SNP between parental lines. *TraesCS2A01G573200* encodes a succinate dehydrogenase subunit; the expression of this gene in Jimai 22 was significantly higher than the susceptible parent, which could serve as a candidate for the map-based cloning of *YrJ22*.

## 4. Materials and Methods

### 4.1. Plant Materials

The wheat variety Jimai 22 showed high resistance to CYR32 (IT = 0~0;), whereas Avocet S was highly susceptible (IT = 4). Jimai 22 was crossed with Avocet S, producing F_1_ and F_2_ (2273 plants) for genetic analysis, and the highly susceptible variety Mingxian169 was used as control. A total of 1022 EMS-induced lines (M_3_) from Jimai 22 were employed for Mutmap-based cloning, and were kindly donated by Drs. Jianmin Song, at the Crop Research Institute, Shandong Academy of Agricultural Sciences, and Xianchun Xia, at the Institute of Crop Sciences, National Wheat Improvement Center, Chinese Academy of Agricultural Sciences (CAAS).

### 4.2. Evaluation of Stripe-Rust Reaction

The stripe-rust inoculations were conducted in the greenhouse of the National Wheat Improvement Center, CAAS. The Jimai 22, Avocet S, 15 F_1_ and 2273 F_2_ plants were scored for stripe-rust resistance by sowing in plastic pots (9 × 9 × 9 cm) with 15 plants each, with three plants of Mingxian 169 as control in each pot. A total of 1022 M_3_ lines (5 seeds per line) were challenged by *Pst* race CYR32 to identify homozygous susceptible-lines. Seedlings at the two-leaf stage were inoculated with CYR32 by brushing fresh conidia from sporulating leaves. Inoculated plants were incubated in a dark dew-chamber for 24 h, with a temperature of 8–12 °C and 100% relative humidity. After incubation, the plants were kept under long-day conditions (a 16-h light/8-h darkness photoperiod, at 15 ± 2 °C) [45]. Approximately 2 weeks after inoculation, when the susceptible control Mingxian169 was fully infected, the infection type (IT) was scored on a 0–4 scale [65,66]. Plants with ITs 0–2 were considered as resistant, and those with 3–4 as susceptible.

### 4.3. BSR-Seq Analysis

The total RNAs were extracted from leaves with an Illumina HiSeq platform at Beijing Novogene Bioinformatics Technology Co. Ltd. Two RNA bulks were prepared by mixing RNA equally from 50 highly resistant (IT = 0;) F_2_ plants and 50 highly susceptible (IT = 4) ones. The RNA quality and concentration was checked using Agilent 2100 Bioanalyzer (Agilent Technologies, Palo Alto, CA, USA) and 1% agarose gel. The quality of the original RNA-Seq reading was evaluated with the software Trimmomatic v0.32 [67]. The low-quality bases and adapter sequences in the raw reads were filtered out to obtain clean reads and a high quality of data. To identity and sort the unique reads with high quality, the alignments between the clean reads and the reference genome-sequence (CS RefSeq v1.0, http://www.wheatgenome.org/, accessed on 12 August 2021) were checked using STAR v2.5.1b [68]. Before the SNPs detection, Samtools rmdup was used to remove PCR duplicates from the PCR fragments with adapters during the library preparation; PCR optical duplicates occurred during clustering in a flow-cell before sequencing [69]. The unique and confident alignments were used to identify SNP variation, using the “Haplotype Caller” module in the GATK v4.0.12.0 software [70]. The SNP variations with allele-frequency difference (AFD) ≥0.9 were regarded as associated with disease resistance, and used as templates for developing SNP markers.

### 4.4. MutMap Analysis

To obtain resequencing data, DNA for MutMap analysis was extracted from the fresh leaves of susceptible M_3_ plants and Jimai 22. Jimai 22 (wild type, WT) and a bulked DNA pool from the equal mixing of DNA from 20 highly susceptible (IT = 4) M_3_ plants, were used for MutMap analysis. The DNA quality and concentration were determined using agarose gel and a DS-11 spectrophotometer (DeNovix, Wilmington, DE, USA). Whole DNA were sequenced using the Illumina HiSeq 2500 platform (Illumina, San Diego, CA, USA), based on the Illumina protocol, including quality control and base-calling analysis by Illumina Casava v1.8. The phred score (Qphred) was used to filter the raw data, to remove the adapter and low-quality base. The clean reads (618.27 Gb, Q30 ≥ 80%, Ave-depth = 20.50×) were aligned to the wheat reference-genome IWGSC RefSeq v1.0 using Burrows Wheeler Aligner (BWA) software. The sequencing reads were mapped to the reference genome and then subjected to subsequent mutation-analysis. The BWA software was mainly used for the alignment of short sequences obtained from second-generation high-throughput sequencing (Illumina HiSeq 2500 sequencing platform) with the reference genome. Over 99.42% reads in both samples were mapped to the reference genome. To identify reliable SNPs by GATK, SAMtools software was used to mark the duplicates first [71]. The reads from the WT and S-bulks were then aligned, and variants were called for in both samples against the developed assembly. The SNP-index (at a position) was defined as the ratio of the count of alternate bases and the count of reads aligned. The SNP-index-plot regression lines were obtained by averaging the SNP indices from a moving window of the 4 Mb interval, with 10 kb increments [72]. The SNP-index can be scanned across the genome to find the region with a value ranging from 0.9 to 1, harboring the gene responsible for the mutant phenotype. The effects of SNPs obtained from association analysis were annotated and predicted, using SnpEff [73].

### 4.5. Genomic DNA Isolation

Genomic DNA was extracted using a high-salt and low-PH reagent [74], in which PVP-40 combines with phenols to prevent DNA browning, and potassium acetate at a low PH allows proteins to precipitate more efficiently.

### 4.6. Development of Polymorphic CAPS and STS Markers

The polymorphic SNPs obtained by BSR-Seq and MutMap analysis were used to design specific primers with Primer Premier5 and WatCut (http://watcut.uwaterloo.ca/template.php, accessed on 25 October 2021). To develop cleaved amplified polymorphic sequences (CAPS) markers for identifying candidate SNP variations, the flanking sequences of each SNP were extended to approximately 2 kb on both sides, based on the CS RefSeq (https://urgi.versailles.inra.fr/blast/blast.php, accessed on 14 September 2021), and were then used as a template [75]. The sequence-tagged-site (STS)-marker loci in the region of the target gene were generated by the BSR-Seq and MutMap by Primer Premier5, following [76].

### 4.7. Quantitative Real-Time PCR Analysis

The total RNA was extracted from the primary leaves of Jimai 22 and Avocet S at 12 and 24 h after inoculation with *Pst* CYR32 with three biological replicates, using a RNAprep Pure Plant Plus Kit (TIANGEN, Beijing, China). First-strand cDNA was synthesized using the HiScript II Q RT Super Mix for qPCR (Vazyme Biotech Co., Ltd., Nanjing, China), and was performed with AceQ Universal SYBR qPCR Master Mix (Vazyme Biotech Co., Ltd., Nanjing, China) in a CFX384TM Real-Time System (Bio-Rad Laboratories, Monza, Italy). The expression of the candidate gene (*TraesCS2A01G573200*) was analyzed using the forward primer 5′-CCGTGGTGCCTCAAACTC-3′ and the reverse primer 5′-GCTCAACTGTGGCTTCTTCA-3′. Three repeats for each RNA sample were run as technical replicates. The wheat actin gene (AB181991) was amplified as the reference gene. Relative expression was determined using the 2^−∆∆ct^ method [77].

### 4.8. Statical Analysis and Genetic Map Construction

Chi-squared (χ^2^) tests were used to determine whether the segregating ratio of F_2_ population fit that expected by SPSS Statistics V22.0. The genotypic data obtained from the polymorphic markers were used for linkage analysis using Mapmaker 3.0, and a genetic map was drawn with the MapDraw V2.1 software [47].

## 5. Conclusions

Jimai 22 is a high-yield cultivar with ASR to stripe rust, and has been frequently used as a parent in wheat breeding. The resistance to CYR32 in Jimai22 was conferred by a new gene, *YrJ2*, on 2AL. *YrJ22* can be pyramided with other *Yr* genes, especially APR genes, to widen its resistance spectrum and enhance a sustainable resistance. The co-segregating marker H732 can be used in MAS for the improvement of stripe-rust resistance in wheat breeding.

## Figures and Tables

**Figure 1 plants-11-03244-f001:**
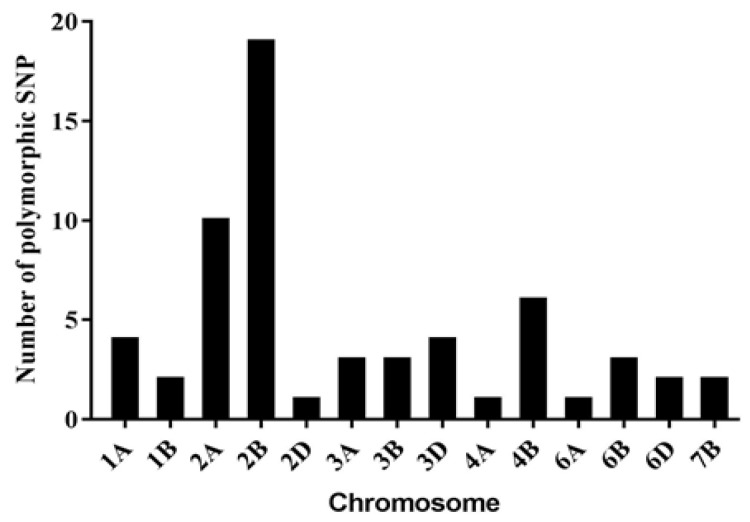
Number of polymorphic SNPs distributing on different chromosomes associated with stripe-rust resistance, based on RNA-Seq of resistant and susceptible bulks.

**Figure 2 plants-11-03244-f002:**
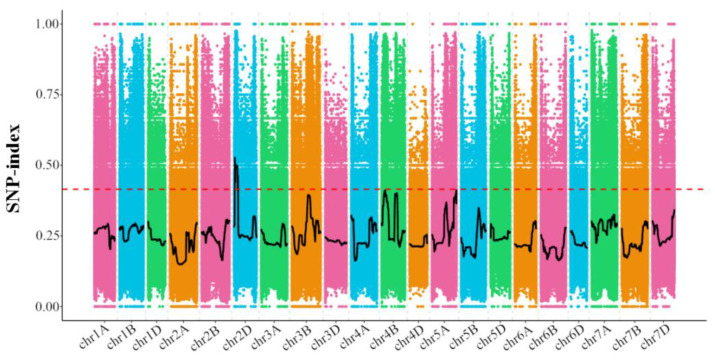
Distribution of SNP-index values on wheat chromosomes using MutMap. SNP-index = 0 means bulked DNA representing wild-type genome. Color dots represent SNP-index by calculation, black line is SNP-index plot. Regression lines were obtained by averaging SNP indices from a moving window of the 4 Mb interval with 10 kb increments. Red line refers to theoretical correlation-threshold line.

**Figure 3 plants-11-03244-f003:**
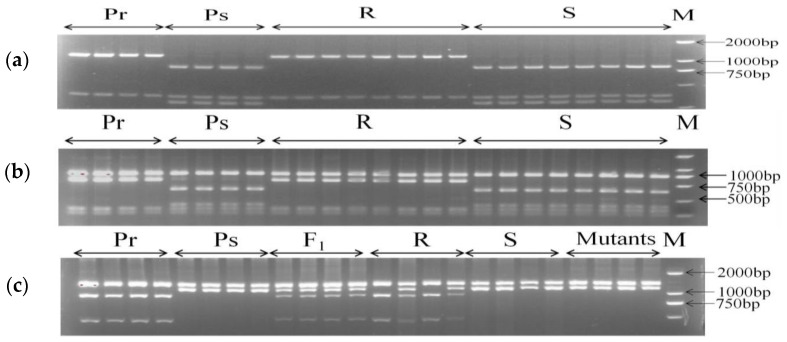
PCR amplification of polymorphic markers (**a**–**c**). Polymorphic test of CAPS markers H400, H732 and H736, respectively, on 1.5% agarose gel. Pr: Jimai 22, Ps: Avocet S, F_1_: F_1_ plants, R: resistant F_2_ plants, S: susceptible F_2_ plants, M: *Trans* 2K DNA Marker.

**Figure 4 plants-11-03244-f004:**
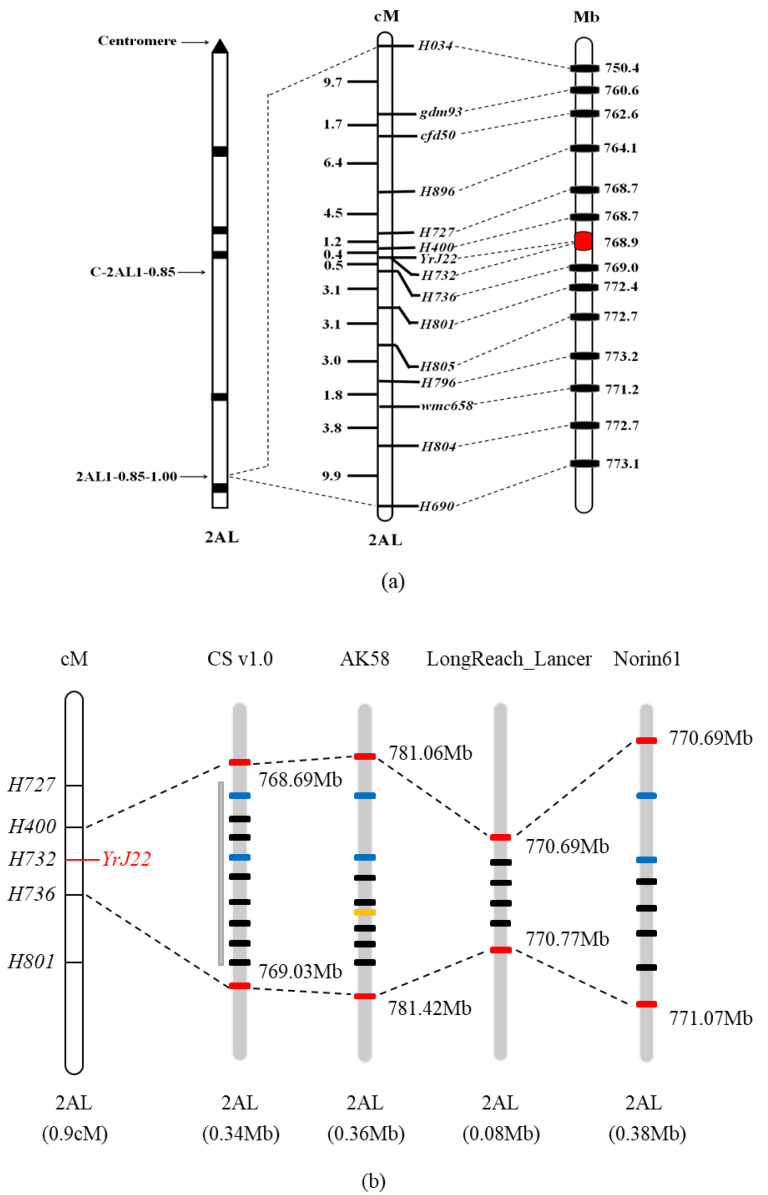
(**a**) Linkage and physical maps for stripe-rust-resistance gene *YrJ22* on chromosome arm 2AL. Loci names are indicated on the right side of the map. Kosambi map distances (cM) are shown on the left side. The physical positions refer to Chinese Spring RefSeq v1.0 (http://www.wheatgenome.org/, accessed on 20 October 2021). (**b**) Collinearity analysis of the target genomic-interval on chromosome 2AL in Chinese Spring RefSeq v1.0 with three *Triticum* species. The picture shows collinear interval of three *Triticum* species flanked by *H727* and *H736*, marked as red rectangles. The blue ones show the *R*-gene within the region. The yellow one indicates the position of novel genes in the target interval of the three species.

**Figure 5 plants-11-03244-f005:**
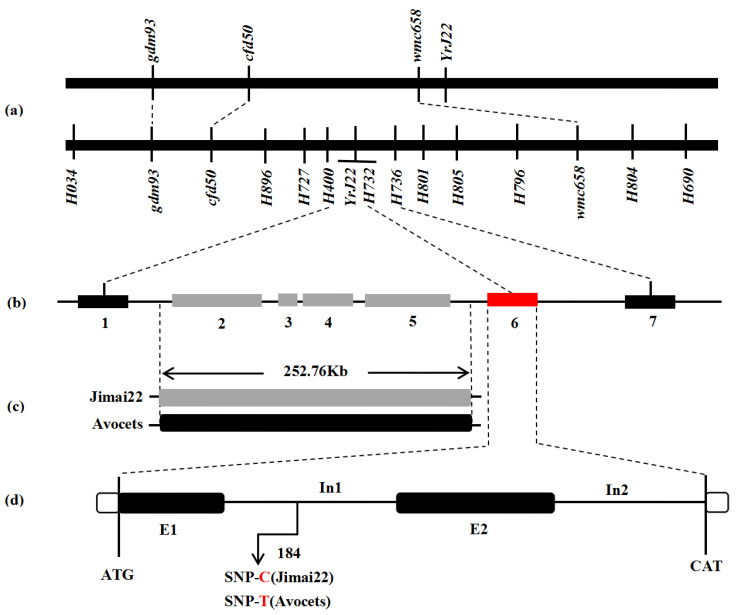
Map-based cloning of the stripe-rust-resistance gene *YrJ22.* (**a**) Genetic maps from a previous study [45] and the present work (lower). (**b**) Physical map of seven polymorphism genes between Jimai 22 and Avocet S. 1: *TraesCS2A01G5740000*, 2: *TraesCS2A01G572900*, 3: *TraesCS2A01G5740100*, 4: *TraesCS2A01G573000*, 5: *TraesCS2A01G573100*, 6: *TraesCS2A01G573200*, 7: *TraesCS2A01G573600.* (**c**) Black box represents region that could be amplified in Avocet S and susceptible lines; grey box indicates the region that is absent in Jimai 22. (**d**) Genomic structure of *TraesCS2A01G573200* in reference sequences of CS. Abbreviations include exon (E) and intron (In).

**Figure 6 plants-11-03244-f006:**
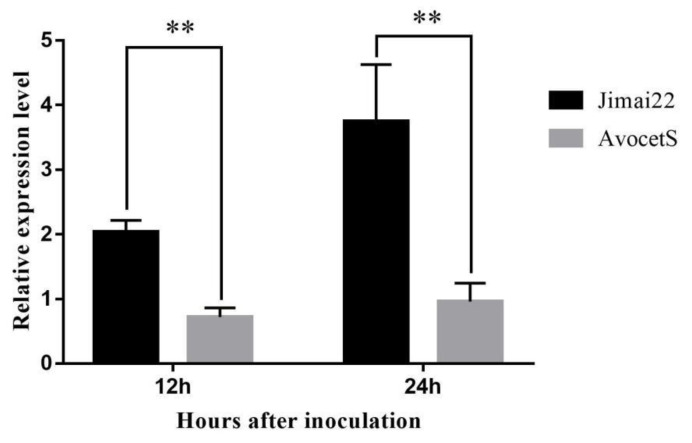
qPCR analysis of the candidate gene *TraesCS2A01G573200* in Jimai 22 and Avocet S at 12 and 24 h after inoculation with CYR32. **, significant difference at *p* < 0.01, based on Fisher’s least significant difference method.

**Table 1 plants-11-03244-t001:** Stripe-rust reactions to *Pst* race CYR32 in parent F_1_ and F_2_ plants.

Plant Material	Total	Infection Type						Expected Ratio	χ^2^	*p*
		0	0	1	2	3	4			
Jimai 22	10	10								
Avocet S	10					3	7			
F_1_	15	15								
F_2_	2273	3	1482	169	66	128	425	3:1	0.46	0.55

**Table 2 plants-11-03244-t002:** Markers developed for mapping *YrJ22*.

Marker	Forward Primer Sequence (5′–3′)	Reverse Primer Sequence (5′–3′)	Physical Pos.* (Mb)	Restriction Enzyme	Product Size (bp)
H796	AGTGGGGCTATTTTGTCGG	CCTTGTTCATGGCTGGTTG	773.2	BSSHII	739
H896	TCTAGTTGCTGCCGTGGTTA	AACCAAACGGACACACATGG	764.1	HpyCH4V	718
H804	GTGCCGTGAGCACCCTGCTG	TGGACGTCTTTTGCCCTCTTGAA	772.6	FoKI	869
H690	GGATTCTCACGGTCACTCAA	CTCATTCCCGCAACAGG	773.0	HaeIII	546
H805	GAGCAGCCGGAGGAGTTG	CAGGAGGAGATGGAGAGCAT	772.7	FoKI	1013
H034	AGTAGGGTAAATGGCGAGCAG	GTCCCAAGGAACAAACACG	750.4	Hpy188III	720
H727	GGGTGGTCACATCCAGGTCC	GAACATGCCTCAGAACAATGGA	768.7	TspRI	443
H400	CGATTGCTGCTTTCCTTCAT	ATCCCATCGGTCCCGTGTT	768.7	BbvI	1677
H732	GCATCGCAGCAACACTCG	GGAGACAATGGGCGGTTT	768.9	BstUI	1128
H736	GTGCTCCTTACAGGGAACAAC	ATCCACAGCCGAACCAAA	769.0	HpyCH4III	1573
H801	GGACAAATACAAGGGTTCG	TGTCGTCGGGATTCAAGG	772.4	-	279

* Physical position based on Chinese Spring reference genome (IWGSC v1.0, http://www.wheatgenome.org/ (accessed on 12 August 2021)).

**Table 3 plants-11-03244-t003:** Thirteen genes in the candidate region.

Name	Length (bp)	Position on Chr2A * (bp)	Functional Annotation *
TraesCS2A01G572700	1455	768,663,628–768,665,082	Peroxidase
TraesCS2A01G572800	1625	768,676,637–768,678,261	Peroxidase
TraesCS2A01G740000LC	1200	768,691,235–768,692,434	F-box domain containing protein
TraesCS2A01G572900(R)	5771	768,729,058–768,734,828	Disease resistance protein
TraesCS2A01G740100LC	483	768,882,506–768,882,988	Endonuclease/exonuclease/ phosphatase family protein
TraesCS2A01G573000	1497	768,923,600–768,925,096	Retrovirus-related Pol polyprotein from transposon TNT 1-94
TraesCS2A01G573100(NLR)	5239	768,926,410–768,931,648	Disease resistance protein (TIR-NBS-LRR class)
TraesCS2A01G573200	1655	768,981,429–768,983,083	Succinate dehydrogenase subunit
TraesCS2A01G740200LC	1251	768,992,115–768,993,365	F-box domain containing protein
TraesCS2A01G573300	1463	768,997,070–768,998,532	Peroxidase
TraesCS2A01G573400	1706	769,003,029–769,004,734	Peroxidase
TraesCS2A01G573500	1136	769,020,358–769,021,493	Peroxidase
TraesCS2A01G573600	1129	769,032,902–769,034,030	Peroxidase

* Refer to Chinese Spring reference genome (IWGSC v1.0, http://www.wheatgenome.org/, accessed on 12 August 2021).

## Data Availability

The data presented in this study is available on request from the corresponding author.

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
