# Peer review of "Fine Mapping of Stripe-Rust-Resistance Gene YrJ22 in Common Wheat by BSR-Seq and MutMap-Based Sequencing"

_plants, 2022, doi:10.3390/plants11233244_

Round 1

Reviewer 1 Report

This manuscript is about fine mapping of a stripe rust resistance gene called YrJ22. The article has a scientific merit if published. However, the authors should make major modification in presenting the results and discussing in detail. Some of my comments and questions are as follows:

1. Introduction need modifications in many places 

1. Line 35 --- occurring by Pst, Lines 39-40 not clear and need revision

Line 47, excavate is not appropriate word

Line 63 needs revision

Line 85 mine?, Line 88 not right place.

I did not see how both BSA and Mutmap were used to narrow the QTL region. The QTL region was reduced to 0.9 cM using classical QTL mapping. Did you identify any SNPs within this 0.9 cM using BSA and Mutmap approaches or they mapped in different regions? not clear. 

Line 257, what is gene expanse?

Did you clone 13 genes?

Lines 222- 238 should be in relation to your results, not just literature review.

Line 247, so there are 3 genes, not 2.

Line 250, what is YrZM175? why it is needed here?

Lines 249-250, not clear and needs revision.

Overall, the manuscript is too short even if you used several gene cloning methods.

Reviewer 2 Report

The manuscript describes that identification and fine mapping of YrJ22 for stripe rust resistance by two bi-parent mapping using BSR-seq and Mutmap analysis. YrJ22 was finally located to a 0.9 cM genetic interval, corresponding to a 340.46 kb of physical region (768.7-769.0 Mb), including 13 high-confidence genes 25 based on Chinese Spring reference genome. And the potential candidate gene was confirmed using sequence differences and gene expression analysis.

Frankly speaking, the work in the present study was very interesting and was important for improvement of wheat resistance breeding in future. 

My major comments are:

1.      L108, please add the number of plants in each bulk. How to select the resistant and susceptible plants? Only based on F2 phenotype?

2.      The author identified a candidate gene but it is not the typical NBS-LRR or kinase gene, I think it is possible that the physical region is not only 340kb in the genome of Jimai22. Please perform comparative genome analysis between CS and the other published genome such as 10+ Genome. And add a collinearity map in the main text.

3.      More details are listed in manuscript, please check it.

Reviewer 3 Report

Please check my comments within the manuscript. 
